# Mesenchymal Stem/Stromal Cells in Organ Transplantation

**DOI:** 10.3390/pharmaceutics14040791

**Published:** 2022-04-04

**Authors:** Dayanand Deo, Misty Marchioni, Prakash Rao

**Affiliations:** Personalized Transplant Medicine Institute, 691 Central Avenue, New Providence, NJ 07974, USA; ddeo@njsharingnetwork.org (D.D.); mmarchioni@njsharingnetwork.org (M.M.)

**Keywords:** mesenchymal stem/stromal cells, transplantation, immunomodulation, tissue engineering, regenerative medicine, paracrine effects, decellularization, organoids, transplant tolerance, 3D bioprinting

## Abstract

Organ transplantation is essential and crucial for saving and enhancing the lives of individuals suffering from end-stage organ failure. Major challenges in the medical field include the shortage of organ donors, high rates of organ rejection, and long wait times. To address the current limitations and shortcomings, cellular therapy approaches have been developed using mesenchymal stem/stromal cells (MSC). MSC have been isolated from various sources, have the ability to differentiate to important cell lineages, have anti-inflammatory and immunomodulatory properties, allow immunosuppressive drug minimization, and induce immune tolerance towards the transplanted organ. Additionally, rapid advances in the fields of tissue engineering and regenerative medicine have emerged that focus on either generating new organs and organ sources or maximizing the availability of existing organs. This review gives an overview of the various properties of MSC that have enabled its use as a cellular therapy for organ preservation and transplant. We also highlight emerging fields of tissue engineering and regenerative medicine along with their multiple sub-disciplines, underlining recent advances, widespread clinical applications, and potential impact on the future of tissue and organ transplantation.

## 1. Introduction

Organ transplantation is unquestionably the preferred standard of care for patients with end-stage organ failure. It has been largely reported that organ transplantation not only increases the overall survival of patients experiencing organ failure but also improves the quality of life of these transplant recipients [1]. Although medicine and technology have shown considerable advancement over the past years, organ transplantation still faces substantial hurdles as the number of candidates listed for transplantation has increased dramatically over the years. As per the United Network of Organ Sharing (UNOS), currently there are over 106,000 patients waiting for a transplant. In 2021, there were 41,354 transplants performed using donated organs from 20,401 living and deceased donors combined [2]. This large disparity has led to long wait times and increased mortality with 17 patients dying every day waiting for a life-saving transplant. Additional challenges for organ donation and transplant include ethical concerns, lack of awareness, logistical issues, availability of specialized equipment, and high cost of organ transplantation and post-transplant immunosuppression medication.

In an attempt to overcome donor organ shortage and effects of long-term immunosuppression, the quest for alternative strategies to allogeneic organ transplantation is gaining traction. One of the most widely studied cell types having a broad-ranging clinical potential are mesenchymal stem/stromal cells (MSC). They act as a central building block in the rapidly growing field of tissue engineering. MSC can be grown rapidly in a culture dish, secrete an abundance of growth factors and cytokines through their paracrine mechanisms, and have been pursued for their ability to induce transplant tolerance [3]. Infusion of MSC from both autologous and allogeneic sources as a cellular therapy have been carried out for their ability to reduce the use of immunosuppressive drugs in organ transplant recipients [4].

Recent advances in the field of tissue engineering and regenerative medicine have introduced new methods and techniques to replace and regenerate functional tissues of clinical relevance. Regeneration of tissue substitutes using an extracellular matrix as a biological scaffold and having a simple architecture such as flat two-dimensional or hollow tubular structures has been developed [5]. The cells used to generate these structures are usually taken from the same patient (autologous) to avoid rejection of the transplanted tissue or organ by the patient’s own immune system. However, cells from other sources (allogeneic), or even stem cells, have been used to generate functional tissue substitutes and protect them from rejection by suppressing the host immune response using immunosuppressant drugs [6].

Creating specific organs using tissue engineering techniques starts with the construction of a scaffold made up of biomaterials such as growth factors, organ specific cells, endothelial cells, and stem cells. However, the most favorable scaffold is that from the original organ itself. This scaffold is generated by decellularization of the target organ thus retaining only the extracellular matrix. The decellularization process removes immune cells and offers a scaffold that has a natural physical and mechanical structure. Recellularization of this scaffold with stem cells allows for their growth, maturation, and differentiation, generating a more mature and fully personalized organ that is ready for transplant [7]. Recently, 3D bioprinting technology has been used in tissue engineering and regenerative medicine to print tissues or organs using additive manufacturing techniques. Three-dimensional structures are created by using appropriate bioinks that are combined with cells and growth factors, and adding these materials in a layer-by-layer process on the scaffold.

The present review aims to systematically assess the potential application of MSC in the context of organ transplantation. Evaluation of unique properties of MSC, including their paracrine and immunomodulatory properties, and ability to induce transplant tolerance and minimization of immunomodulatory drugs are noted. Furthermore, advances in the field of tissue engineering and regenerative medicine technologies that use MSC to develop complex 3D structures suitable for transplantation of functional tissues and organs are discussed.

## 2. Properties of MSC

Historically, bone marrow was the first source from which MSC were isolated, showing their plastic adherence property. Reports of isolation of MSC from various developmental stages (including fetal, young, adult and older population) from various sources having similar properties have since been published [8,9]. Sources of fetal tissues from where MSC have been isolated include Wharton’s jelly (umbilical cord), umbilical cord blood, placenta, amniotic fluid, and chorionic villi. Adult sources of MSC include tissues and secretions such as adipose tissue, dental pulp, menstrual blood, peripheral blood, yellow ligament, endometrium, and mother’s milk [10,11,12,13,14,15,16,17,18,19,20,21,22,23] (Figure 1).

We have isolated and characterized MSC from solid adipose tissue obtained from research consented deceased donors. Our results demonstrate that an increased number of MSC can be obtained from deceased donor adipose tissue using our newly developed nonenzymatic technique as compared to the conventional enzymatic method. Deceased donor adipose tissue can be recovered during the routine deceased donor process, substantially increasing the supply and access to MSC without the pain, morbidity, and mortality associated with living donor stem cell collection [24].

The main criterion for the identification of MSC is their ability to grow in vitro as a population adhering to the substrate. These cells are phenotypically characterized by the expression of CD73, CD90, CD105 surface antigens and the lack of expression of CD4, CD34, CD14, CD11b, CD79a, CD19 or class II human leukocyte antigens (HLA II) [25,26]. Some additional markers such as stromal-1 antigen (STRO-1), vascular cell adhesion molecule (VCAM/CD106), and melanoma cell adhesion molecule (MCAM/CD146) were also identified as being useful during the isolation of MSC having multidirectional differentiation ability and high degree of clonogenicity [27,28,29].

It has been observed that the rate of proliferation of MSC differs amongst the different sources from which these cells have been isolated. MSC isolated from fetal origin tissues have a faster rate of proliferation and a greater number of in vitro passages until senescence than those isolated from adult sources [30]. A higher degree of ‘stemness’ is demonstrated by the ability of cells to create a large number of CFU-F colonies. This is a characteristic of MSC isolated from bone marrow and adipose tissues [31,32]. MSC isolated from umbilical cord blood lack the ability to differentiate into adipocytes whereas bone marrow and adipose tissue MSC have a greater tendency to differentiate to osteoblasts [33]. Age of the donor also plays an important part in the proliferation ability of MSC. A greater percentage of apoptotic cells, slower rate of proliferation, and weaker differentiation potential have been observed when MSC have been isolated from older donors [34,35,36].

The morphology of cultured MSC isolated from the same tissue can be differentiated into three sub-populations: (i) cells resembling fibroblasts with characteristic spindle-shaped proliferating cells, (ii) multi-granular large flat cells with a clearly marked cytoskeleton, and (iii) small round cells having self-renewal capacity [37]. Thus, the morphology of MSC suggests that these cells are multipotent having multi-directional differentiation potential which is the most critical characteristic of MSC. The differentiation of MSC into different end-stage lineage cells highly depends on the tissue source. Bone-marrow-derived MSC have the greatest capacity to differentiate into the three mesenchymal cell lineages (osteoblast, chondrocyte, and adipocyte) [38]. On the other hand, adipose-derived MSC have been shown to differentiate into cardiomyocytes, hepatocytes, and islet cells [39,40,41]. Umbilical cord blood-derived MSC have a biological advantage over MSC isolated from other tissues. They have the ability of large-scale expansion, high anti-inflammatory effects, significant retardation of senescence, and capacity of longer culture times. These properties enable umbilical cord blood MSC to differentiate into neuronal cells, cardiomyocytes, endothelial cells, skeletal cells, and the three mesenchymal cell lineages (osteoblast, chondrocyte, and adipocyte). MSC from liver have a strong differentiation potential towards chondrogenic and osteogenic lineages, but less towards adipogenic lineages. MSC isolated from Wharton’s jelly have been successfully differentiated into endothelial cells after the addition of vascular endothelial growth factor (VEGF) and skeletal muscle lineages such as bone, cartilage, and skeletal muscle cells [42,43,44]. Thus, selecting the source of MSC for clinical transplantation depends on the capacity of MSC to differentiate towards certain lineages that is relevant to the specific transplantation site. We have shown the ability of adipose-derived MSC, obtained from research consented deceased donors by a novel nonenzymatic technique, to successfully differentiate into beta cells. These cells demonstrate secretion of c-peptide and insulin under glucose challenge thus confirming their functionality. The availability of an abundant supply of adipose-derived MSC from deceased donors and their subsequent differentiation into functional beta cells renders them as a promising cellular therapeutic approach for treating patients with type 1 diabetes (T1D) [45].

## 3. Functions of MSC in Organ Transplantation

MSC have unique functional properties that have been used to enable better engraftment and acceptance of the transplanted organ (Figure 2).

### 3.1. Paracrine Signaling

The resolution of adult tissue damage is a complex process that is not likely resolved by MSC alone. Harnessing the ability of MSC to produce factors and cytokines that can modulate immune responses and inflammation, as well as stimulate tissue repair, has been the focus of many recent studies. Clinical trials are ongoing to test how to utilize the paracrine activity of MSC in addition to the ability to differentiate to mesenchymal lineages. The paracrine secretions of MSC, commonly referred to as ‘secretome’ have the ability to support regenerative processes in damaged tissues. The secreted factors have a distinct impact on various regulatory processes [46,47]. The major components that make up the MSC secretome include growth factors and cytokines. Some of the growth factors secreted by MSC include VEGF, bFGF, NGF, TGF, and KGF [48,49,50]. During the process of regeneration, these growth factors help in the reduction of fibrosis of tissues. Growth factors secreted by MSC having an anti-apoptotic effect include HGF, CICN-3, TIMP-1, TIMP-2, and BDNF [51,52,53,54]. Factors in the secretome that stimulate the proliferation of cells include NGF, bFGF, IGFBP-1, and M-CSF. Chemokines secreted by MSC have the function of blocking or stimulating cell chemotaxis and include CCL5 (RANTES), CCL8 (MCP-2), and CXCL12 (SDF-1) [55]. Factors promoting angiogenesis in the secretome include VEGF, whereas anti-angiogenesis factors such as IFN are also present within the secretome. MSC secretome also contains factors that have anti-bacterial, anti-viral, and anti-parasitic activity [56,57,58,59]. It has been shown that some of the factors within the MSC secretome have anti-cancer activity and interact with cancer cells to reduce their proliferation, migration, and viability [60]. We have identified various cytokines, growth factors, and signaling proteins in the supernatant of MSC grown in tissue culture flasks. Maximum secretion of these factors was observed on day 4 of MSC culture that decreased by day 6 culture. The paracrine secretions of MSC were also observed in the secretome that separates as a byproduct during the isolation of adipose-derived MSC. Further studies are needed to evaluate the levels of these factors secreted among different donor age groups and their ability to enhance the tissue repair functionality of MSC (recent, unpublished observations).

In recent years, the paracrine activity of MSC has expanded to include a rapidly developing field of secreted extracellular vesicles (EVs). These include apoptotic bodies, microvesicles, and exosomes. These EVs play an important role in the regulation of immune response, homeostasis, and other biological functions. The EVs have a composition that coincides with the cellular components of MSC from which they originate [61,62,63]. Supernatants from the in vitro culture of MSC have been shown to protect other cells from apoptosis, have immunomodulatory effects, induce proliferation of cells, stimulate angiogenesis, prevent fibrosis of tissues, and induce stem cell differentiation [64,65,66].

### 3.2. Immunomodulation

MSC have a major advantage in regulating the immune response due to their ability to secrete various regulatory factors. All these multiple factors acting in unison enable the immune-modulatory and anti-inflammatory properties of MSC thus acting as a master regulator of the immune system. Both primary and acquired immune responses are regulated by the interaction of MSC with various effector cells [67]. The anti-inflammatory properties of MSC include blocking the differentiation of CD34^+^ cells into mature dendritic cells using secreted factors as well as direct contact of MSC with dendritic cells [68], therefore limiting the mobilization of the mature dendritic cells to the tissues [69]. T-cell proliferation is inhibited by IL-10 that is secreted by MSC while the pro-inflammatory M1 macrophages are transformed into anti-inflammatory M2 macrophages phenotype by interaction with MSC [70]. In vitro studies have shown that MSC suppress the activation of CD4^+^ T cells and CD8^+^ B cells [71], reduce levels of pro-inflammatory TNF and IFN cytokines, and increase the synthesis of anti-inflammatory IL-4 cytokines [72]. MSC are able to skew the maturing immune cell populations by increasing the levels of regulatory T cells (Treg), anti-inflammatory Th2 cells, and dendritic DC2 cells, while reducing the levels of pro-inflammatory Th1 cells, DC1 cells, and NK cells [73]. The synthesis of immunoglobulins such as IgM, IgG, and IgA that are secreted by B cells and their differentiation to plasma cells can be blocked by MSC, reducing the levels of circulating antibodies. The ability of B cells to migrate is also negatively affected by MSC as they can reduce the expression of chemokine receptors on the surface of B cells [74].

Among the immunomodulatory properties, MSC are able to block apoptosis of activated neutrophils, limit their mobilization to the area of damage, and reduce their binding to vascular endothelial cells [75]. MSC also inhibit the complement-mediated effects of peripheral blood mononuclear cell proliferation [76]. Pro-inflammatory cytokines secreted by MSC are able to limit mast cell degranulation and migration towards chemotactic factors and stimulate neutrophil chemotaxis [77,78]. MSC also block the proliferation of induced NK cells and aid in the reduction of the cytotoxic activity of NK cells [79].

### 3.3. Transplant Tolerance

Organ transplantation is the standard of care for end-stage organ failure patients [80]. While conventional immunosuppression protocols have improved short-term outcomes of transplant recipients, the long-term outcomes are not as favorable, leading to chronic rejection and death. In addition, prolonged immunosuppression leads to an increase in side effects such as malignancies, infections, toxicities, and other metabolic diseases. The survival rate for a functioning heart, liver, and kidney is approximately 50%, while that for lung is only 30%, ten years post-transplantation [81]. The most desirable outcome of transplantation, often described as the “Holy Grail” of transplantation, is the establishment of transplant tolerance. Tolerance in the setting of organ and tissue transplantation would abrogate the need for chronic immunosuppression, eliminate drug-related side effects, and extend organ half-life thereby addressing the critical shortage of organ availability for transplant [82].

After organ or tissue transplantation, interplay between the host immune system and the transplanted organ has led to the development of T-cell immunosuppressive agents which has abated the risk of acute rejection. To achieve long-term graft survival, the identification of innovative strategies that lead to allograft tolerance should be developed. MSC are a heterogeneous population of non-hematopoietic cells that are able to differentiate into tissues of mesodermal lineages. In the pursuit of transplant tolerance induction, MSC seem a very promising cellular therapy for the minimization or even discontinuation of lifelong immunosuppression. MSC can dampen the activation of cells of both the adaptive and innate immune systems, reprogramming them into regulatory cells inhibiting the immune alloresponse at different levels. Infusion of MSC has proven to be effective in prolonging graft survival and controlling autoimmunity [83]. MSC reduce the host-vs-graft response by secreting soluble immunomodulatory factors through paracrine mechanisms and through contact-dependent regulation [84,85] tipping the balance of alloresponse from effector to regulatory function. Therefore, acute and chronic alloimmune response can be reduced using MSC that not only target T cells which are the main players of alloimmunity, but also regulate B cells, dendritic cells, and macrophages.

Autologous and allogeneic MSC have been shown to regulate the T-cell response to the transplanted organ by suppressing T-cell cytotoxicity and proliferation caused by antigen priming and polyclonal activators [86]. The prevalence of a high frequency of alloreactive memory T cells before transplantation is a barrier to tolerance induction, especially in the context of T-cell depletion therapy, and has a deleterious effect on allograft survival. Human autologous and allogeneic MSC have been shown to suppress and inhibit these memory T cells, including CD8^+^ memory T cells, leading to allograft survival [87,88]. MSC are able to modulate the activity of helper T cells by suppressing the proliferation of Th1 cells and rewiring their polarization to the favorable Th2 cells. This shift in the polarization of helper T cells also helps in an increase in the secretion of IL-10 and other Th2 specific cytokines [89]. In addition to the suppressing effector T cells, MSC have been shown to expand a pool of regulatory T cells thus modulating the immunological response towards the transplanted organ. Human MSC promote the generation of FoxP3^+^ regulatory T cells (T_REG_) and induce tolerance to the allograft by mediating immunosuppression in vivo [85,90].

B cells are known to secrete antibodies that bind to the donor organ leading to antibody-mediated rejection (AMR). MSC have been shown to inhibit the formation of donor-specific antibodies (DSA) [91] and block B-cell proliferation through cell cycle arrest [74]. Inhibition of the proliferation of B cells by MSC prevents them from maturing into plasmablasts causing a steep decline in antibody secretion [92]. MSC infusion modulates the humoral response in vivo by expanding the number of regulatory B cells (B_REG_) along with IL-10 production [93]. Spontaneous operational tolerance has been shown in kidney allograft recipients receiving MSC infusion through the increase in ‘transitional’ B-cell subsets [62,94].

Dendritic cells (DC) are responsible for the direct alloantigen presentation to CD4^+^ T cells and cross-presentation of allopeptides to CD8^+^ T cells. Exposure to MSC interferes with DC proliferation, maturation, impairs DC homing to secondary lymphoid organs, and downregulates MHC Class II and costimulatory molecules [95]. This has a direct effect on alloantigen presentation on both recipient and donor DC leading to the inhibition of alloresponse and the induction of regulatory phenotype by increasing the abundance of T_REG_ compared to effector T cells. Kidney transplant recipients receiving MSC infusion showed an increased suppressive T_REG_ population in the graft and secondary lymphoid organs, impaired donor-specific T-cell proliferation, and a high frequency of immature tolerogenic DC [96].

Macrophage proliferation and migration have been shown to increase in the presence of MSC along with a pro-tolerogenic shift in polarization of the macrophages to the M2 phenotype [97]. In a corneal transplant model, infusion of MSC conferred protection towards allograft rejection by redirecting the macrophages towards the M2 phenotype [98]. The macrophage M2 is a pro-tolerogenic phenotype that reduces TNF, IFN, and IL-12 secretion and increases the production of IL-10 that promotes T_REG_ proliferation and inhibits effector T-cell responses [70].

### 3.4. Timing of MSC Infusion to Develop Transplant Tolerance

The lifespan of MSC is limited both in tissue culture system in vitro and after in vivo administration raising the question of the optimal timing for MSC infusion in transplant recipients. Development of long-term tolerance to the transplanted organ after a single infusion of MSC would only be possible if multilevel protolerogenic effects could be developed in the short period of time the MSC are actively engaging with their in vivo environment [62,99]. MSC infusion in rats carried out for four days before heart transplantation induced transplant tolerance and acceptance of the transplanted heart. However, rejection of the transplanted heart was observed when MSC infusion was performed three days after performing the heart transplant [63,100]. Timing of MSC infusion may also impact their localization. MSC have been shown to migrate to the graft rather than to the secondary lymphoid organs after transplantation. This difference in the localization dictates the immunomodulatory properties of MSC. Migration of MSC to the transplanted organ after post-transplant infusion leads to the stimulation of proinflammatory phenotype characterized by complement deposition, neutrophil infiltration, and graft rejection. On the other hand, pre-transplant infusion leads to the localization of MSC to the secondary lymphoid organs promoting the protolerogenic effects and prolonged graft survival [101]. Furthermore, the infusion of MSC post-transplant was unable to convert conventional T cells to the immunoregulatory T_REG_ population and reduce the inhibitory effect of MSC on DC maturation, therefore decreasing the immunomodulatory ability of MSC [102].

### 3.5. Minimization of Immunosuppressive Drugs

Development of a pro-tolerogenic environment and repair of chronic allograft damage by MSC therapy in kidney transplantation has enabled the minimization of induction and maintenance immunosuppressive drugs [103,104]. A clinical trial showed that induction therapy using basiliximab could be replaced safely with the administration of BM-MSC infusion, resulting in a 50% reduction of tacrolimus used as maintenance therapy. Patients receiving MSC therapy displayed incidence of acute rejection, graft survival and function, similar to the patients receiving a full dose of tacrolimus maintenance immunosuppression [105,106]. In another clinical trial, the timing of BM-MSC infusion was shown to be critical in living donor kidney transplant recipients. Patients were given an autologous MSC infusion a day before kidney transplantation. At first, basiliximab was removed from the induction therapy to avoid inhibition of T_REG_ expansion by this drug. There was no acute renal insufficiency, but the absence of this drug led to an increased risk of acute rejection. The protocol was modified and basiliximab induction therapy was re-introduced in the pre-transplant MSC-infused patients. The 5–7-year follow-up showed stable graft function with no major side effects. Extensive longitudinal immunological studies in these patients displayed a long-lasting increase in the levels of T_REG_ cells compared to the CD8^+^ T cells and persistent reduction of T-cell cytotoxicity. In one of the patients showing no evidence of de novo donor-specific antibodies and normal histology at 1-year post-transplant, gradual tapering and withdrawal of immunosuppressive drugs cyclosporin and mycophenolate mofetil led to the patient being free from immunosuppression and stable graft function [103,104,106]. Thus, MSC therapy has shown to create a pro-tolerogenic environment and complement the tolerogenic potential of induction therapies for prevention of acute graft rejection. Identification of MSC therapy response biomarkers would aid in the selection of patients who are amenable to safe immunosuppressive drug withdrawal.

## 4. Regenerative Approach as ‘Bridge to Transplant’

The field of transplant medicine has seen major advances in surgical techniques and immunosuppressive therapy saving lives through organ and tissue transplants. However, according to the Health Resources and Services Administration (HRSA), there is a major imbalance between organ supply and demand. Over 100,000 people are on the national transplant waitlist; with a patient being added to this list every 9 minutes, while 17 people die each day while waiting for a life-saving transplant. Of the limited number of eligible donors, less than 1% of all deaths meet the specific medical criteria to be a donor [107]. Additional factors posing a barrier include a narrow timeframe of organ viability after procurement, inadequate organ preservation/storage, and complex transplant logistics. In the past decade, the fields of regenerative medicine (RM) and tissue engineering have emerged offering different approaches and novel strategies as a ‘bridge to transplant’ to overcome these hurdles and tackle unsolved problems [108]. Transplant medicine has benefited a great deal from the various aspects of RM as they offer flexibility and planning of transplantation, lower immunogenicity, generation of universally accepted ‘off-the-shelf organs-on-demand’, increase in organ utilization, expansion of the donor pool, reduction in disparities, and enabling faster access to transplantation [109] (Figure 3).

### 4.1. Organ Preservation

Soon after organ procurement, the lack of blood supply leads to oxygen deprivation, electrolyte imbalance, anaerobic respiration, and metabolic waste accumulation leading to ischemic injury. Reperfusion of the organ leads to epithelial cell damage and generation of reactive oxygen species (ROS), aggravating the already damaged organ called reperfusion injury. Prolonged ex vivo preservation of organs and tissues has been made possible by the development of machine perfusion techniques. Organ perfusion using an oxygenated cold preservation solution (hypothermic oxygenated perfusion) focuses on slowing down metabolic rates [110,111], whereas ex vivo perfusion at physiological temperature mimics the in vivo environment (normothermic perfusion) [112]. Both these approaches allow for the measurement of predictive biomarkers including but not limited to neutrophil gelatinase-associated lipocalin (NGAL), liver-type fatty acid binding protein 1, kidney injury molecule-1 (KIM-1), endothelin-1, and (L-FABP) micro ribonucleic acids (miRNAs) that give valuable information of organ quality [113]. Machine perfusion offers a perfect platform for the delivery of treatments solely to the graft, bypassing complications arising from systemic delivery. This is beneficial for ECD and DCD donation, facilitating organ storage, transportation, tissue repair, and organ regeneration [114]. Delivery of stem cells using machine perfusion as a treatment option is gaining importance. Several preclinical studies are being conducted to investigate the effect of paracrine factors secreted by MSC and MSC-derived extracellular vesicles in kidney and liver perfusion [115,116]. Results demonstrated that there was significantly less global ischemic damage in DCD rat kidneys that were perfused with MSC or MSC-derived extracellular vesicles [117,118].

### 4.2. Decellularization and Recellularization

The process of isolation of the extracellular matrix (ECM) from any given tissue or organ maximizing the removal of cellular debris with minimal loss, damage, or disruption to the tissue or organ, is known as decellularization. DNA content in the decellularized tissue is reduced to less than 4% while structural proteins such as collagen, laminin, and fibronectin remain in the scaffold [119,120]. The decellularized scaffold can influence cell migration, proliferation, and differentiation, and therefore serves as more than just a framework for cells [121]. The use of decellularized scaffolds for organ bioengineering has focused on organs such as heart, lungs, kidney, liver, and pancreas [122]. Sufficient decellularization is essential to reduce immunogenicity [123]. The final goal of decellularization is to ultimately repopulate the scaffold with patient-specific pluripotent cells to generate a personalized ‘organ-on-demand’. This process of recellularization necessitates seeding of terminally differentiated somatic cells or stem cells onto the scaffold via perfusion of cells through the vasculature, ureter, or trachea, and further maturation in special bioreactors [124,125,126]. Neo-bladders were generated by Atala et al. [127] by first decellularized bladder submucosa followed by recellularization with donor-derived muscle and urothelial cells. After ex vivo culture, these bioengineered bladders were implanted in patients who showed no postoperative complications and normal bladder function. Recellularization of a whole organ remains a challenge due to the need of seeding and precise positioning of different cell types, thromboembolic events during perfusion, inadequate supply of oxygen and nutrients, and improper vascularization of recellularized organs [122]. Preclinical studies in mice have shown promising results. Decellularized mouse hearts that were repopulated with induced pluripotent stem cell (iPSC)-derived cardiovascular cells exhibited spontaneous contractions after 20 days [128]. Repopulation of lung scaffolds with lung epithelial cells and vascular endothelial cells showed survival and differentiation of the epithelium and clearance of secretions [126]. Bioengineered mouse kidneys prepared after decellularization followed by recellularization with human umbilical vein endothelial cells (HUVEC), neonatal kidney cells, and mouse embryonic stem cells, displayed increased creatinine clearance, albumin retention, and increased reabsorption of glucose and electrolyte [125]. Seeding of hepatic stem cells into decellularized liver matrix exhibited markers of hepatocyte and cholangiocytic differentiation and high engraftment rates [129].

### 4.3. Organoids

Most two-dimensional studies carried out in vitro fail to replicate the in vivo interactions among cells and between cells and the extracellular matrix. The development of 3D culture systems allowed the mimicking of in vivo conditions involving interactions between cells and the surrounding matrix leading to the dynamic regulation of signaling pathways and paracrine signals [130]. Organoids are 3D structures created in this culture system typically originating from stem cells having multiple cell types that self-organize in culture [131]. They are miniaturized version of organs having native architecture and morphology, and display several biological interactions that occur in vivo [132]. Several controlled parameters are used to induce iPSCs to differentiate into specific lineages to form tissue specific organoids. These include endogenous and exogenous signals that stimulate the differentiation of iPSCs and self-organization into the 3D structures. Organoids can also be developed by seeding differentiated stem cells along with endothelial cells and MSC in combination to form self-assembled 3D structures [133]. At present, organoid technology is used for drug screening and disease modeling. However, the ultimate goal is to evaluate the transplantation of tissue-specific organoids and access their engraftment, biocompatibility, and tissue specific functionality in vivo. Most of the studies on the efficacy of transplantation of organoids is carried out using nude mouse model. Kidney organoids derived from human pluripotent stem cell (hPSC) differentiation were transplanted under the renal capsule of immunodeficient mice. These organoids exhibited glomerular vascularization and connection with preexisting host vascular networks, functional glomerular perfusion, maturation of podocytes, and tubular reabsorption [134].

Immune deficiency caused by the destruction of pancreatic beta cells leads to the development of type 1 diabetes (T1D). Pancreatic organoids have been developed by differentiation of hPSC into acinar and ductal cells, and inducing them to self-organize into pancreatic organoids. These organoids can express pancreatic markers and are functionally and structurally similar to the pancreas. Pancreatic organoids were placed in a 3D-printed tissue trapper and implanted into the peritoneal cavity of immunodeficient mice. Results indicated that the implanted organoids exhibited engraftment, neo-vascularization, an increased number of insulin-positive cells, and improved c-peptide secretion suggesting their applicability in the treatment of T1D [135].

Liver organoids were formed by inducing the coculture of hPSC, MSC, and HUVEC to self-organize into 3D structures resembling liver buds. These liver organoids, when transplanted into nude mice, displayed functional vascularization, connections among donor and host cells, and drug metabolism activity; all essential functional components of liver function [133]. However, there are practical challenges to the use of organoids for transplantation purposes. Organoids are typically 10 mm to 1 mm in diameter. The small size of organoids is a major concern for their use as a substitute for larger organs such as the kidney. Production of bigger organoids developed using more precursor cells or assembling large number of organoids could possibly alleviate this issue [136]. Most of the studies rely on transplantation studies carried out in immunodeficient mouse model. Translation of these studies into human trials to establish the use of organoids as medical devices to replace or improve organ function is far from reality at present. Use of larger humanized animal models for preclinical studies could help overcome this limitation [137]. Tracking the ongoing in vivo engraftment, vascularization, behavior, and function of transplanted organoids is an important aspect of organoid transplantation technology. The use of iPSC expression fluorescent biosensors could help in creating an informative tracking system [138].

### 4.4. 3D Bioprinting

One of the latest approaches to meet the increasing demand of organs for critical human organ transplantation is the use of 3D bioprinting technology. Bioprinting is an additive fabrication process that uses layer-by-layer addition of living cells and growth factors suspended in an appropriate bioink to create three-dimensional structures [139]. There are three basic steps in the 3D bioprinting process: model the 3D structure of tissue or organ using a computer modelling program, printing the 3D structure using bioink, and post-processing assessment of the physical, mechanical, and biological functions before transplanting into patients. Recent advances in 3D bioprinting have the potential of meeting the demands of tissues and organs for transplant [140]. The critical components of the 3D bioprinting process are the selection of a suitable bioink and appropriate cell type selection. The bioink material should provide appropriate growth and adhesion factors, signaling proteins, and mechanical and structural properties of the extracellular cell matrix [141]. Cell selection for 3D organ bioprinting involves the selection of cells possessing proliferation and differentiation capacity in the printed scaffold. These cells should interact with signaling molecules and be able to survive and remain viable during and after the printing process [142]. Recent advances in the field of 3D bioprinting have gained importance as this process can improve the quality of life and also save lives of patients. Burn victims are greatly benefited by the use of 3D bioprinting technology through the development of 3D-printed artificial skin that contains multiple cell types such as keratinocytes, melanocytes, and fibroblasts layered on a desired scaffold [143]. Restoration of damaged cartilage has been possible by incorporating 3D printing technology using cartilaginous tissue scaffolds with chondrocytes, MSC, and bone marrow cells [144]. Using a CT scan of the patient, 3D-printed bone implants using MSC-differentiated osteoblasts were designed to perfectly fit the broken bone fragment [144]. Three-dimensional bioprinting of the heart involves the initial CT scan of the donor’s heart followed by using stem cells, growth factors, and appropriate bioink mixed with hydrogel. The scaffold that is used to layer the bioink provides mechanical support and the exact shape of the patient’s heart where the cells grow and proliferate. The synchronous beating of the cells is soon observed and the scaffold is then ready for implantation [145].

## 5. Conclusions and Future Perspectives

MSC have been recognized as a major player in the field of transplant medicine as they play an active role in homeostasis of tissues and organs. They are the driving force behind developing new technologies in biomedical research as they display their therapeutic effects through their ability to differentiate into different tissue types and stimulate the regeneration of damaged tissues. Although MSC compose a negligible fraction of cells in vivo, they can be isolated from various tissues and body fluids and subject to in vitro expansion and storage, known in clinical terms as biomanufacturing and biobanking. The critical importance of paracrine effects of MSC are now being recognized and advocated to explain their functional benefits. MSC secrete many different growth factors and cytokines that have immunomodulatory properties, regulate inflammation and migration of cells to the damaged tissue, influence wound healing, promote angiogenesis, and protect host cells from apoptosis. These complex paracrine mechanisms lead to improved quality and functionality of tissue repair. MSC cell therapy is especially impactful for transplant patients as they shift the risk:benefit ratio by inducing immune tolerance thus alleviating the burden of lifelong immunosuppression, associated morbidity, and chronic rejection, significantly improving transplant outcomes.

Maximizing the use of available tissue and organs has been achieved by focusing on new technologies in tissue engineering and regenerative medicine. Fabrication of new organs by optimizing discarded donor organs as a source of organ scaffold using machine perfusion and decellularization techniques is gaining importance. Recellularization of tissue or organ scaffolds using 3D printing technology employing sequential layering of a mixture of lineage-specific differentiated cells and appropriate bioink has shown promising results. Three-dimensional culture systems have been developed that allow self-aggregation of MSC, endothelial cells, and terminally differentiated cells to produce organoids that reproduce the structural complexity of a real organ. Organoids have differentiated and functional cells capable of interacting with host cells. Organoids have also been used as promising tools for drug screening, disease modeling, and personalized medicine thus reducing whole organ transplant requirement.

In summary, it appears that MSC are becoming a powerful tool in the field of transplant medicine by increasing organ preservation, utilization, and immune tolerance. Increase in the scientific knowledge and clinical applications of MSC is beginning to contribute towards the transformation of the therapeutic applications in organ and tissue transplantation. Major discoveries in the tissue engineering and regenerative medicine fields, originally thought as ‘illusions’ or ‘science fiction’, have enabled significant advances of truly disruptive approaches and technologies that could forever change the field of transplantation as we know it today. It would be prudent to speculate the next milestone in tissue engineering and regenerative medicine: the transplantation of an artificial, bioengineered functioning whole organ in humans.

## Figures and Tables

**Figure 1 pharmaceutics-14-00791-f001:**
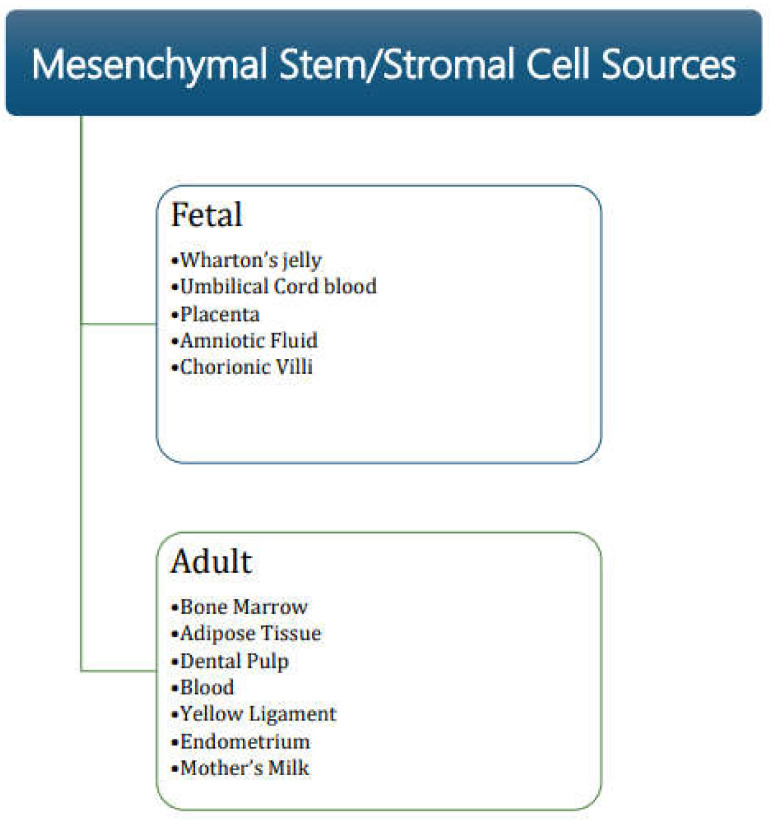
Sources of mesenchymal stromal/stem cells.

**Figure 2 pharmaceutics-14-00791-f002:**
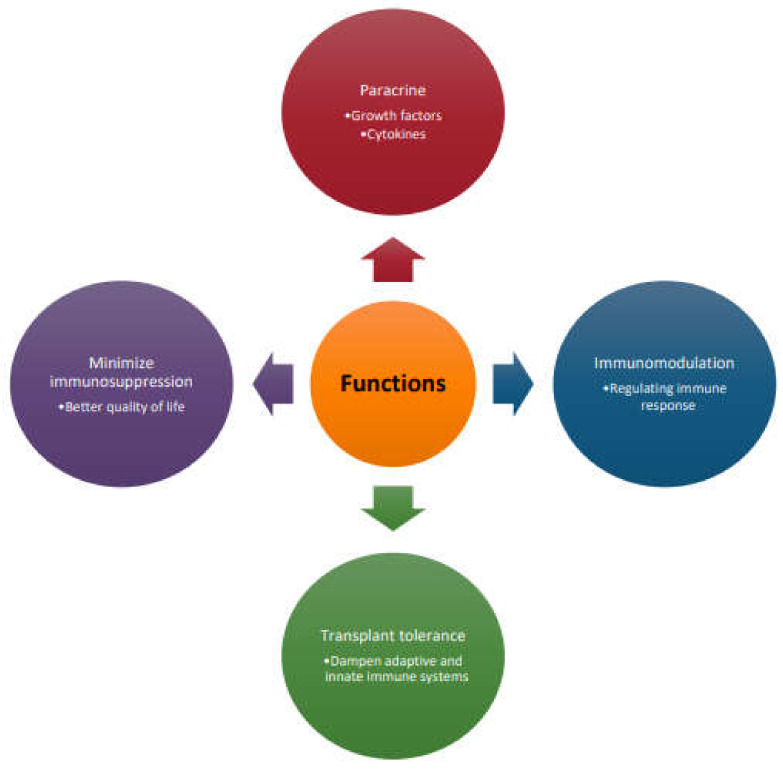
Functional properties of MSC that enable better engraftment of the transplanted organ.

**Figure 3 pharmaceutics-14-00791-f003:**
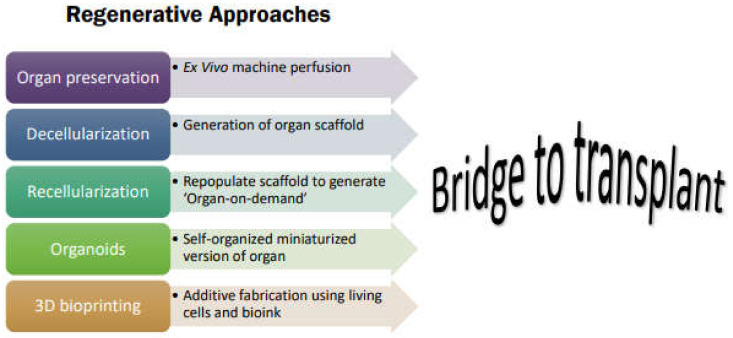
Schematic representation of novel technologies in tissue engineering and regenerative medicine using MSC as a bridge to organ transplantation.

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
