# Peer review of "Mesenchymal Stem/Stromal Cells in Organ Transplantation"

_pharmaceutics, 2022, doi:10.3390/pharmaceutics14040791_

Round 1

Reviewer 1 Report

The authors comprehensively organize the immunomodulatory properties of mesenchymal stem/stromal cells (MSC), and give some examples to illustrate that the administration of MSCs before or after organ transplantation will have different effects on transplant immune tolerance. It was also observed in clinical trials that MSC infusion can reduce the dosage of immunosuppressive drugs. The authors gave a detailed description of the application of MSCs from the induction of transplant tolerance to the application of tissue engineering and regenerative medicine. If there is a picture to summarize the essence of this article, it will make it easier for readers to understand.

Author Response

We would like to thank you for reviewing our review article titled “Mesenchymal Stem/Stromal Cells in Organ Transplantation”. We have addressed the concerns raised by the reviewers and have included additional information as per their suggestions. The changes made have been marked up using the “Track Changes” function of MS Word in the manuscript. We are submitting the revised manuscript for favor of review and publication.

The details of the revisions are as follows:

  1. Reviewer’s comment: “If there is a picture to summarize the essence of this article, it will make it easier for readers to understand.”

Response:

We thank the reviewer for the detailed analysis of our manuscript and the valuable suggestion of including a summary picture to help the reader better understand the gist of the review article. We had added the following figures to the manuscript.

  • Figure 1: Sources of mesenchymal stromal/stem cells
  • Figure 2: Functional properties of MSC that enable better engraftment of the transplanted organ
  • Figure 3: Schematic representation of novel technologies in tissue engineering and regenerative medicine using MSC as a bridge to organ transplantation

Reviewer 2 Report

Dayanand Deo, Prakash Rao and colleagues systematically assess the potential application of MSCs in organ transplantation. The manuscript briefly describes the unique properties of MSCs, including their paracrine and immunomodulatory properties and their ability to induce transplant tolerance and minimize immunomodulatory drugs. In the manuscript, the authors detail advances in tissue engineering and regenerative medicine techniques using MSCs to develop complex 3D structures suitable for functional tissue and organ transplantation, including decellularization, organoids, and 3D bioprinting.

Some additional comments are given below:

  1. The references are relatively old, and authors are advised to add the most cutting-edge articles and progress in various fields to enrich the content of the review.
  2. Please double check and correct typos in the manuscript. For example, Page 9, line 19.
  3. There are no figures in the manuscript. It is recommended that the author add some figures to help readers in different fields understand.

Author Response

We would like to thank you for reviewing our review article titled “Mesenchymal Stem/Stromal Cells in Organ Transplantation”. We have addressed the concerns raised by the reviewers and have included additional information as per their suggestions. The changes made have been marked up using the “Track Changes” function of MS Word in the manuscript. We are submitting the revised manuscript for favor of review and publication.

The details of the revisions are as follows:

  1. Reviewer’s comment: “The references are relatively old, and authors are advised to add the most cutting-edge articles and progress in various fields to enrich the content of the review.”

Response:

We have added new references that represent current strategies to overcome hurdles in organ transplantation using tissue engineering and regenerative medicine. The new references are highlighted in the ‘Reference’ section of the manuscript and the corresponding number of the new references are added to the appropriate sequence in the manuscript.

  1. Reviewer’s comment: “Please double check and correct typos in the manuscript. For example, Page 9, line 19.”

Response:

We thank the reviewer for the critical analysis of the manuscript and noting the typographical errors that were overlooked by the authors. Upon thorough review of the manuscript, we found additional places that needed edits to provide better clarity. All these edits have been shown as mark-ups in the ‘Track Changes’ function of MS Word.

  1. Reviewer’s comment: “There are no figures in the manuscript. It is recommended that the author add some figures to help readers in different fields understand.”

Response:

We appreciate this valuable suggestion and agree that the addition of figures representing the essence of the manuscript would help readers in related and unrelated fields better understand the importance of the application of MSC in organ transplantation. We had added the following figures to the manuscript.

  • Figure 1: Sources of mesenchymal stromal/stem cells
  • Figure 2: Functional properties of MSC that enable better engraftment of the transplanted organ
  • Figure 3: Schematic representation of novel technologies in tissue engineering and regenerative medicine using MSC as a bridge to organ transplantation

Reviewer 3 Report

The review was aimed to give an overview of the various properties of the cellular therapy with Mesenchymal Stem Cells for organ preservation and transplant.

My comments are:

Generally, the references in the text are appropriate, but sometimes it would be necessary to extend the bibliography or upgrade it with some recent entries.

For instance, I would suggest that you add:

- near the ref 31

“Saler M, Caliogna L, Botta L, Benazzo F, Riva F, Gastaldi G. hASC and DFAT, Multipotent Stem Cells for Regenerative Medicine: A Comparison of Their Potential Differentiation In Vitro. Int J Mol Sci. 2017 Dec 13;18(12):2699. doi: 10.3390/ijms18122699. PMID: 29236047; PMCID: PMC5751300”

- near the ref 34

“Piñeiro-Ramil M, Sanjurjo-Rodríguez C, Rodríguez-Fernández S, Castro-Viñuelas R, Hermida-Gómez T, Blanco-García FJ, Fuentes-Boquete I, Díaz-Prado S. Generation of Mesenchymal Cell Lines Derived from Aged Donors. Int J Mol Sci. 2021 Oct 1;22(19):10667. doi: 10.3390/ijms221910667. PMID: 34639008; PMCID: PMC8508916.”

- near the ref 49

“Xu R, Feng Z, Wang FS. Mesenchymal stem cell treatment for COVID-19. EBioMedicine. 2022 Mar 10;77:103920. doi: 10.1016/j.ebiom.2022.103920. Epub ahead of print. PMID: 35279630; PMCID: PMC8907937.”

“Navard SH, Rezvan H, Haddad MHF, Ali SA, Nourian A, Eslaminejad MB, Behmanesh MA. Therapeutic effects of mesenchymal stem cells on cutaneous leishmaniasis lesions caused by Leishmania major. J Glob Antimicrob Resist. 2020 Dec;23:243-250. doi: 10.1016/j.jgar.2020.09.005. Epub 2020 Sep 23. PMID: 32977079.”

and

“Harrell CR, Popovska Jovicic B, Djonov V, Volarevic V. Molecular Mechanisms Responsible for Mesenchymal Stem Cell-Based Treatment of Viral Diseases. Pathogens. 2021 Apr 1;10(4):409. doi: 10.3390/pathogens10040409. PMID: 33915728; PMCID: PMC8066286.”

-near the ref 75

“De Martino M, Zonta S, Rampino T, Gregorini M, Frassoni F, Piotti G, Bedino G, Cobianchi L, Dal Canton A, Dionigi P, Alessiani M. Mesenchymal stem cells infusion prevents acute cellular rejection in rat kidney transplantation. Transplant Proc. 2010 May;42(4):1331-5. doi: 10.1016/j.transproceed.2010.03.079. PMID: 20534294.”

-near the ref 73

“Gregorini M, Bosio F, Rocca C, Corradetti V, Valsania T, Pattonieri EF, Esposito P, Bedino G, Collesi C, Libetta C, Frassoni F, Dal Canton A, Rampino T. Mesenchymal stromal cells reset the scatter factor system and cytokine network in experimental kidney transplantation. BMC Immunol. 2014 Oct 3;15:44. doi: 10.1186/s12865-014-0044-1. PMID: 25277788; PMCID: PMC4193986.”

- near the ref 80

“Kaundal U, Ramachandran R, Arora A, Kenwar DB, Sharma RR, Nada R, Minz M, Jha V, Rakha A. Mesenchymal Stromal Cells Mediate Clinically Unpromising but Favourable Immune Responses in Kidney Transplant Patients. Stem Cells Int. 2022 Feb 15;2022:2154544. doi: 10.1155/2022/2154544. PMID: 35211176; PMCID: PMC8863486.”

- near the ref 99

“Gregorini M, Corradetti V, Pattonieri EF, Rocca C, Milanesi S, Peloso A, Canevari S, De Cecco L, Dugo M, Avanzini MA, Mantelli M, Maestri M, Esposito P, Bruno S, Libetta C, Dal Canton A, Rampino T. Perfusion of isolated rat kidney with Mesenchymal Stromal Cells/Extracellular Vesicles prevents ischaemic injury. J Cell Mol Med. 2017 Dec;21(12):3381-3393. doi: 10.1111/jcmm.13249. Epub 2017 Jun 21. PMID: 28639291; PMCID: PMC5706569.”

- near the ref 100

“Aubin H, Kranz A, Hülsmann J, Lichtenberg A, Akhyari P. Decellularized whole heart for bioartificial heart. Methods Mol Biol. 2013;1036:163-78. doi: 10.1007/978-1-62703-511-8_14. PMID: 23807795.”

- near the ref 110

“Shah SB, Singh A. Cellular self-assembly and biomaterials-based organoid models of development and diseases. Acta Biomater. 2017 Apr 15;53:29-45. doi: 10.1016/j.actbio.2017.01.075. Epub 2017 Jan 31. PMID: 28159716.”

The authours are invited to increase the references concerning the MSC secretome ( paragraph <Paracrine signaling> ). Example can be

  • near the ref 45

“Han T, Song P, Wu Z, Xiang X, Liu Y, Wang Y, Fang H, Niu Y, Shen C. MSC secreted extracellular vesicles carrying TGF-beta upregulate Smad 6 expression and promote the regrowth of neurons in spinal cord injured rats. Stem Cell Rev Rep. 2021 Aug 27. doi: 10.1007/s12015-021-10219-6. Epub ahead of print. PMID: 34449013.” ;

“Paganelli A, Trubiani O, Diomede F, Pisciotta A, Paganelli R. Immunomodulating Profile of Dental Mesenchymal Stromal Cells: A Comprehensive Overview. Front Oral Health. 2021 Mar 31;2:635055. doi: 10.3389/froh.2021.635055. PMID: 35047993; PMCID: PMC8757776.”

and

 “Shin S, Lee J, Kwon Y, Park KS, Jeong JH, Choi SJ, Bang SI, Chang JW, Lee C. Comparative Proteomic Analysis of the Mesenchymal Stem Cells Secretome from Adipose, Bone Marrow, Placenta and Wharton's Jelly. Int J Mol Sci. 2021 Jan 15;22(2):845. doi: 10.3390/ijms22020845. PMID: 33467726; PMCID: PMC7829982.”

  • -near the ref 46-47

“Gonzalez-Rey E, Gonzalez MA, Varela N, O'Valle F, Hernandez-Cortes P, Rico L, Büscher D, Delgado M. Human adipose-derived mesenchymal stem cells reduce inflammatory and T cell responses and induce regulatory T cells in vitro in rheumatoid arthritis. Ann Rheum Dis. 2010 Jan;69(1):241-8. doi: 10.1136/ard.2008.101881. PMID: 19124525.”

and

“Wang P, Cui Y, Wang J, Liu D, Tian Y, Liu K, Wang X, Liu L, He Y, Pei Y, Li L, Sun L, Zhu Z, Chang D, Jia J, You H. Mesenchymal stem cells protect against acetaminophen hepatotoxicity by secreting regenerative cytokine hepatocyte growth factor. Stem Cell Res Ther. 2022 Mar 4;13(1):94. doi: 10.1186/s13287-022-02754-x. PMID: 35246254; PMCID: PMC8895877.”

The ref 51 is inappropriate, perhaps it could be replaced by

 “van Balkom BWM, Gremmels H, Giebel B, Lim SK. Proteomic Signature of Mesenchymal Stromal Cell-Derived Small Extracellular Vesicles. Proteomics. 2019 Jan;19(1-2):e1800163. doi: 10.1002/pmic.201800163. Epub 2019 Jan 4. PMID: 30467989.”

The most recent references are missing in the paragraph <Timing of MSC infusion to develop transplant tolerance>. I would suggest you these articles:

“Kaundal U, Ramachandran R, Arora A, Kenwar DB, Sharma RR, Nada R, Minz M, Jha V, Rakha A. Mesenchymal Stromal Cells Mediate Clinically Unpromising but Favourable Immune Responses in Kidney Transplant Patients. Stem Cells Int. 2022 Feb 15;2022:2154544. doi: 10.1155/2022/2154544. PMID: 35211176; PMCID: PMC8863486.”

and

“Casiraghi F, Todeschini M, Azzollini N, Cravedi P, Cassis P, Solini S, Fiori S, Rota C, Karachi A, Carrara C, Noris M, Perico N, Remuzzi G. Effect of Timing and Complement Receptor Antagonism on Intragraft Recruitment and Protolerogenic Effects of Mesenchymal Stromal Cells in Murine Kidney Transplantation. Transplantation. 2019 Jun;103(6):1121-1130. doi: 10.1097/TP.0000000000002611. PMID: 30801518; PMCID: PMC6934941.”

In the paragraph <Organ preservation> the authors report the different approaches and novel strategies that could be taken to minimize the gap between organ supply and demand. The authors do not report in the text, the organ preservation through hypothermic oxygenated machine

perfusion or the delivery of MSC-derived extracellular vesicles during the machine perfusion.

You can consider a few suggestions:

“Dondossola D, Ravaioli M, Lonati C, Maroni L, Pini A, Accardo C, Germinario G, Antonelli B, Odaldi F, Zanella A, Siniscalchi A, Cescon M, Rossi G. The Role of Ex Situ Hypothermic Oxygenated Machine Perfusion and Cold Preservation Time in Extended Criteria Donation After Circulatory Death and Donation After Brain Death. Liver Transpl. 2021 Aug;27(8):1130-1143. doi: 10.1002/lt.26067. PMID: 33835695.”

“Ravaioli M, De Pace V, Angeletti A, Comai G, Vasuri F, Baldassarre M, Maroni L, Odaldi F, Fallani G, Caraceni P, Germinario G, Donadei C, Malvi D, Del Gaudio M, Bertuzzo VR, Siniscalchi A, Ranieri VM, D'Errico A, Pasquinelli G, Morelli MC, Pinna AD, Cescon M, La Manna G. Hypothermic Oxygenated New Machine Perfusion System in Liver and Kidney Transplantation of Extended Criteria Donors: First Italian Clinical Trial. Sci Rep. 2020 Apr 8;10(1):6063. doi: 10.1038/s41598-020-62979-9. Erratum in: Sci Rep. 2020 Sep 1;10(1):14658. PMID: 32269237; PMCID: PMC7142134.”

“De Stefano N, Navarro-Tableros V, Roggio D, Calleri A, Rigo F, David E, Gambella A, Bassino D, Amoroso A, Patrono D, Camussi G, Romagnoli R. Human liver stem cell-derived extracellular vesicles reduce injury in a model of normothermic machine perfusion of rat livers previously exposed to a prolonged warm ischemia. Transpl Int. 2021 Sep;34(9):1607-1617. doi: 10.1111/tri.13980. PMID: 34448268.”

and

“Rampino T, Gregorini M, Germinario G, Pattonieri EF, Erasmi F, Grignano MA, Bruno S, Alomari E, Bettati S, Asti A, Ramus M, De Amici M, Testa G, Bruno S, Ceccarelli G, Serpieri N, Libetta C, Sepe V, Blasevich F, Odaldi F, Maroni L, Vasuri F, La Manna G, Ravaioli M. Extracellular Vesicles Derived from Mesenchymal Stromal Cells Delivered during Hypothermic Oxygenated Machine Perfusion Repair Ischemic/Reperfusion Damage of Kidneys from Extended Criteria Donors. Biology. 2022; 11(3):350. https://doi.org/10.3390/biology11030350”

Author Response

We would like to thank you for reviewing our review article titled “Mesenchymal Stem/Stromal Cells in Organ Transplantation”. We have addressed the concerns raised by the reviewers and have included additional information as per their suggestions. The changes made have been marked up using the “Track Changes” function of MS Word in the manuscript. We are submitting the revised manuscript for favor of review and publication.

The details of the revisions are as follows:

  1. Reviewer’s comment: “Generally, the references in the text are appropriate, but sometimes it would be necessary to extend the bibliography or upgrade it with some recent entries.”

Response:

We thank the reviewer for the suggestion to extend the bibliography and for the list of suggested references. The suggestions were very appropriate and fit well within the manuscript. We have included the suggested references in the manuscript.  The new references are highlighted in the ‘Reference’ section of the manuscript and the corresponding number of the new references are added to the appropriate sequence in the manuscript.

  1. Reviewer’s comment: “The authours are invited to increase the references concerning the MSC secretome (paragraph <Paracrine signaling>).”

Response:

We appreciate the reviewer’s concern for expanding references concerning the MSC secretome. We have included additional references that pertain to MSC secretome.  The new references are highlighted in the ‘Reference’ section of the manuscript and the corresponding number of the new references are added to the appropriate sequence in the manuscript.

  1. Reviewer’s comment: “The ref 51 is inappropriate, perhaps it could be replaced by “van Balkom BWM, Gremmels H, Giebel B, Lim SK. Proteomic Signature of Mesenchymal Stromal Cell-Derived Small Extracellular Vesicles. Proteomics. 2019 Jan;19(1-2):e1800163. doi: 10.1002/pmic.201800163. Epub 2019 Jan 4. PMID: 30467989.”

Response:

We agree that the “van Balkom BWM, Gremmels H, Giebel B, Lim SK. Proteomic Signature of Mesenchymal Stromal Cell-Derived Small Extracellular Vesicles. Proteomics. 2019 Jan;19(1-2):e1800163. doi: 10.1002/pmic.201800163. Epub 2019 Jan 4. PMID: 30467989” is more relevant than the previous reference used and have replaced Reference 51 with the suggested reference.

  1. Reviewer’s comment: “The most recent references are missing in the paragraph <Timing of MSC infusion to develop transplant tolerance>.”

Response:

We thank the reviewer for the suggestion of including recently published articles pertaining to the timing of MSC infusion to develop transplant tolerance. We have added these references to the manuscript. The new references are highlighted in the ‘Reference’ section of the manuscript and the corresponding number of the new references are added to the appropriate sequence in the manuscript.

  1. Reviewer’s comment: “In the paragraph <Organ preservation> the authors report the different approaches and novel strategies that could be taken to minimize the gap between organ supply and demand. The authors do not report in the text, the organ preservation through hypothermic oxygenated machine perfusion or the delivery of MSC-derived extracellular vesicles during the machine perfusion.”

Response:

Although the application of hypothermic machine perfusion has been mentioned in the ‘Organ preservation’ section of the manuscript, the reviewer’s suggestion of including hypothermic ‘oxygenated’ machine perfusion is noted and the necessary edits have been made to the manuscript. These edits have been shown as mark-ups in the ‘Track Changes’ function of MS Word and the suggested references have been added to the manuscript. Preclinical studies using application of MSC-derived extracellular vesicles in kidney and liver perfusion has been mentioned towards the end of the paragraph under ‘Organ preservation. The new references suggested by the reviewer have been included and are highlighted in the ‘Reference’ section of the manuscript. The corresponding number of the new references are added to the appropriate sequence in the manuscript.

Round 2

Reviewer 2 Report

Authors have improved the manuscript significantly in regards to the original version and added new references and figures that represent current strategies to overcome hurdles in organ transplantation using tissue engineering and regenerative medicine. I think the novel reference can help readers more. From my point of view, the new version of the manuscript that is including the answers to my questions and inquiries has been well performed. For me, this new version is including all indicated aspects for considering to be integrated.

Author Response

Reviewer II:

  1. Reviewer’s comment: “Authors have improved the manuscript significantly in regards to the original version and added new references and figures that represent current strategies to overcome hurdles in organ transplantation using tissue engineering and regenerative medicine. I think the novel reference can help readers more. From my point of view, the new version of the manuscript that is including the answers to my questions and inquiries has been well performed. For me, this new version is including all indicated aspects for considering to be integrated.”

Response:

We thank the reviewer for the critical review of the manuscript. We are encouraged by the reviewer’s comments that the manuscript has been significantly improved after incorporating the suggestions of adding novel references and figures. We greatly appreciate the reviewer’s support in considering the manuscript for publication in the journal Pharmaceutics.